# Orthohantavirus Replication in the Context of Innate Immunity

**DOI:** 10.3390/v15051130

**Published:** 2023-05-09

**Authors:** Autumn LaPointe, Michael Gale, Alison M. Kell

**Affiliations:** 1Department of Molecular Genetics and Microbiology, University of New Mexico, 915 Camino de Salud NE, Albuquerque, NM 87131, USA; 2Department of Immunology, Center for Innate Immunity and Immune Disease, University of Washington, Seattle, WA 98109, USA

**Keywords:** orthohantavirus, viral replication, innate immune sensing, antiviral response

## Abstract

Orthohantaviruses are rodent-borne, negative-sense RNA viruses that are capable of causing severe vascular disease in humans. Over the course of viral evolution, these viruses have tailored their replication cycles in such a way as to avoid and/or antagonize host innate immune responses. In the rodent reservoir, this results in life long asymptomatic infections. However, in hosts other than its co-evolved reservoir, the mechanisms for subduing the innate immune response may be less efficient or absent, potentially leading to disease and/or viral clearance. In the case of human orthohantavirus infection, the interaction of the innate immune response with viral replication is thought to give rise to severe vascular disease. The orthohantavirus field has made significant advancements in understanding how these viruses replicate and interact with host innate immune responses since their identification by Dr. Ho Wang Lee and colleagues in 1976. Therefore, the purpose of this review, as part of this special issue dedicated to Dr. Lee, was to summarize the current knowledge of orthohantavirus replication, how viral replication activates innate immunity, and how the host antiviral response, in turn, impacts viral replication.

## 1. Introduction

Orthohantaviruses are tri-segmented, negative sense, single-stranded RNA viruses that cause significant and severe vascular disease. These viruses are maintained as persistent, asymptomatic infections in rodent and insectivore reservoirs around the world [1,2,3]. Human orthohantavirus infection occurs when contaminated rodent feces are aerosolized and inhaled. Orthohantavirus disease manifests as hemorrhagic fever with renal syndrome (HFRS) or hantavirus cardiopulmonary syndrome (HCPS) [4,5,6,7,8]. Caused by orthohantaviruses found predominately in Asia, Russia, and Eastern Europe (described as Old World viruses), HFRS severity spans from febrile illness to hemorrhagic fever and kidney disease and is associated with a mortality rate of up to 12% [4,8,9]. HCPS, associated with orthohantaviruses found in the Americas (New World viruses), initially presents as mild fever and myalgia but can escalate to acute respiratory failure with a mortality rate of up to 60% [4,5,7]. Although the tissue tropism of disease appears to be unique between HFRS and HCPS, symptoms of inflammation, vascular leakage, thrombocytopenia, and coagulopathy are shared [10,11,12,13]. Endothelial cells are the central targets of orthohantavirus infections in reservoirs and humans and, because orthohantaviruses are not cytopathic, orthohantavirus disease is hypothesized to be immune-mediated [14,15,16]. Thus, investigations into how the immune response to orthohantaviruses is initiated and regulated in humans, and how a robust response is dampened in rodent reservoirs, is a central focus of orthohantavirus research.

In human and rodent hosts alike, virus infection is sensed by specific proteins called pathogen recognition receptors (PRRs). In particular, Toll-like receptors (TLRs) and RIG-I like receptors (RLRs) are major PRRs of RNA virus recognition whereby they recognize and bind to viral products known as pathogen associated molecular patterns (PAMPs), thereby triggering their signaling functions to activate downstream transcription factors including interferon regulatory factor (IRF)3 and NF-KB [17,18,19,20,21]. IRF3 and NF-KB direct the immediate expression of target genes encoding antiviral, inflammatory, and immune modulatory proteins including types I and III interferon (IFN), immune cytokines, and chemokines [22]. Many direct IRF3-target genes/proteins, such as the interferon-induced protein with tetratricopeptide repeats (IFIT) family of proteins, have antiviral function and their rapid expression serves to immediately suppress various steps of viral replication [22,23]. IFNs produced from the infected cell then mediate autocrine and paracrine signaling via specific IFN receptors and the Jak-STAT pathway, leading to the induction of hundreds of interferon-stimulated genes (ISGs). ISG products have antiviral, immune modulatory, metabolic, and cell viability functions that establish a tissue-wide and even systemic antiviral state while serving to polarize the adaptive immune response to infection [24]. Virtually all cells express the receptor chains for type I IFN, while type III IFN receptor expression is largely restricted to mucosal sites and specific myeloid cells [24,25]. Innate immune activation, with production and response to both IFN types, is considered important for orthohantavirus restriction [3,26]. As described below, RLRs, including RIG-I and MDA5 and specific TLRs, are implicated as PRRs of orthohantaviruses to direct innate immune activation, but the recognized viral PAMPs, the host cell innate immune effector genes, and the ISGs of antiviral actions against orthohantaviruses are not well defined [27]. Moreover, while orthohantaviruses encode viral proteins that can facilitate the evasion of PRR actions, the linkage between innate immune evasion and the development of HFRS or HCPS is not defined [28]. Furthermore, while innate immune responses are known to be different for pathogenic and nonpathogenic orthohantaviruses, it is unclear what roles they play in determining disease development [29,30,31]. As presented below, we reviewed the state of the field of innate immunity in orthohantavirus infection, replication, and disease, and define specific knowledge gaps for research focus.

When Dr. Ho Wang Lee isolated and characterized the first orthohantavirus, the Hantaan virus (HTNV), he not only ended a decades long search for the etiologic agent for an important disease in the region, but his breakthrough also ushered in a wave of discovery for orthohantaviruses worldwide and established a new field of virology [32,33,34,35,36,37,38]. This review pays homage to that field by presenting what we have learned over the last 47 years about how orthohantavirus replication activates the host immune response and how antiviral immunity inhibits viral replication. We highlighted the significant progress made in understanding orthohantavirus molecular biology and we concluded by emphasizing outstanding gaps in knowledge and new technologies that promise to further unlock the secrets of these viruses, originally discovered by Dr. Lee.

## 2. Viral Replication and Innate Immune Response

### 2.1. Viral Attachment and Entry

Orthohantavirus infection of target endothelial cells begins with attachment of the glycoproteins to receptors on the cell surface. Interestingly, the specific receptor may differ between Old World and New World viruses. β3 integrins were first reported to be an important group of receptors for several orthohantaviruses. Old World viruses, HTNV, Seoul virus (SEOV), and Puumala virus (PUUV), may use αvβ3, while the New World viruses, Sin Nombre (SNV) and New York 1 (NY-1), are proposed to interact with α11β3 during the infection of Vero cells and human umbilical vein endothelial cells (HUVEC) [39,40,41]. A related non-pathogenic New World orthohantavirus, Prospect Hill (PHV), was reported to use α5β1 integrins for entry into both Vero cells and HUVEC [39,40]. Recent studies have identified the cellular protein protocadherin-1 (PCDH1) as necessary for the attachment and entry of New World viruses SNV and Andes (ANDV) into human lung microvascular endothelial cells [42,43]. Underscoring the importance of PCDH1 for New World virus infection and pathogenicity, the otherwise lethal Syrian hamster model was completely protected from ANDV challenge when PCDH1 was genetically knocked out [43]. Curiously, PCDH1 was not required for Old World orthohantavirus (SEOV and HTNV) entry into human lung microvascular endothelial cells, suggesting host and tissue tropism has evolved separately for these virus groups. Further, it is not known whether innate immune activation alters the expression of critical receptors for orthohantavirus infection or whether immune regulation could modify susceptibility to attachment.

Virus internalization occurs through clathrin-mediated endocytosis, although some evidence suggests that ANDV also uses dynamin-independent macropinocytosis [44,45,46,47]. The viral particle is then trafficked to the late endosome and lysosomal compartments where a drop in pH facilitates viral membrane fusion with the endosomal membrane, releasing the viral genome and associated proteins into the cytoplasm [48,49,50]. This process was recently reviewed in depth by Mittler et al. [51].

During viral attachment and entry, the virus may be detected by TLRs. In particular, TLR3, which detects endosomal dsRNA, is reportedly involved in innate immune induction during HTNV infection of the human hepatocyte cell line Huh7 [52]. Interestingly, these researchers also reported that TLR3 in Huh7 was dispensable for innate immune activation during infection by the non-pathogenic PHV. We have reported, however, that TLR3 knockout in murine embryonic fibroblasts (MEF) did not alter HTNV infection kinetics, suggesting that the role of TLR3 in orthohantavirus infection may be cell type and host dependent [27]. In addition, intact TLR3 signaling was not sufficient to rescue ISG expression in HTNV-infected RIG-I-like receptor knockout HUVEC. The role of TLR3 is further complicated by work demonstrating that ectopic TLR3 expression in TLR3-defficient Huh7.5 cells did not facilitate innate immune activation in response to UV-inactivated SNV [53]. Because Huh7.5 cells lack functional RIG-I signaling, this result implies a role for RIG-I, and not TLR3, in innate immune activation in response to SNV in this model [54]. In addition to TLR3, reports have suggested that the activation of TLR4, located on the outer cell membrane and functioning as a PRR for bacterial LPS, may initiate an antiviral response in HTNV infection. TLR4 knockdown in EVC-304 cells resulted in decreased translocation of NF-κB and IRF3 to the nucleus as well as decreased production of IFN-β, IL-6, and TNF-α in response to HTNV infection [55,56]. Although EVC-304 cells were originally identified as an endothelial cell line, they have recently been shown to have more epithelial-like characteristics [57]. While it is currently not understood how TLR4 may detect orthohantavirus infection, recent work has demonstrated TLR4 activation by viral glycoproteins belonging to other viral families, including Dengue, Ebola, SARS-CoV-2, respiratory syncytial, and vesicular stomatitis viruses [58,59,60,61,62,63]. These data suggest that TLR4 may be capable of detecting HTNV glycoproteins during viral attachment to the cell surface, although more research is needed to provide direct evidence of such an interaction.

Notably, TLR expression can vary significantly among endothelial cells, depending on the tissue and host species [64]. For example, in a study comparing TLR expression among multiple endothelial cell types, primary human aortic endothelial cells (HAEC) expressed high levels of TLRs 1 and 4 and low levels of TLRs 5–10, while primary HUVECs expressed high levels of TLRs 1, 2, 4–8, and 10 and low levels of TLR9 [65]. Similarly, a comparative review on the expression of TLRs in different animal species showed that humans and rats differed in their expression of TLR4 in both the umbilical vein endothelium and brain endothelial cells [66,67]. This natural variation of TLR expression may explain differences observed in studies of TLR-driven immune activation during orthohantavirus infection [27,52,53]. Furthermore, differential expression of TLRs could play a role in orthohantavirus tissue tropism, with preferential infection of cells with low TLR abundance. Overall, studies to date have shown that TLRs play a complex role in the immune response to viral infection, and the impact of the cell-specific nature of TLR expression on orthohantavirus infection remains an open avenue for future study.

Paracrine signaling of type I IFN in neighboring, uninfected cells induces robust expression of antiviral interferon-induced transmembrane (IFITM) proteins [68]. IFITM proteins prevent fusion of the viral envelope with the host plasma or endosomal membrane and their antiviral activity has been described for several RNA viruses [69,70,71,72,73,74]. IFITM1, 2, and 3 have reported antiviral activity against Old and New World orthohantaviruses [70,75,76]. Overexpression of IFITM1, 2, or 3 in Vero cells significantly decreased the percentage of cells infected with either ANDV or HTNV 24 h post-infection [70]. In an A549 lung epithelial cell model, knockdown of IFITM1 or IFITM3 resulted in increased HTNV replication, while overexpression of IFITM3 decreased HTNV infection [75]. In HUVEC, only IFITM3 effectively inhibited HTNV replication. A more recent report found that overexpression of IFITM3 in Vero cells did not significantly diminish PUUV titers when compared to controls, suggesting that the antiviral activity of IFITM proteins may vary against different orthohantaviruses [76]. Additionally, whether IFITM proteins similarly restrict nonpathogenic hantaviruses is unknown.

Overall, research to date has shown that the interplay between innate immunity and viral entry is complex. TLR3 and TLR4 have both been reported to be able to detect orthohantavirus infection, but may do so in a cell type dependent manner. Meanwhile, the IFITM proteins have been shown to be able to restrict multiple orthohantaviruses, although the efficiency of their antiviral activity varies between different orthohantaviruses. Some remaining questions concerning viral entry and innate immunity include: (1) whether innate immune activation alters the expression of the cellular receptors for orthohantavirus attachment and entry; (2) if immune regulation could modify susceptibility to attachment; (3) and how the cell-specific nature of TLR expression impacts orthohantavirus infection.

### 2.2. Viral RNA Replication and Transcription

Fusion of the viral envelope to the endosomal membrane releases the viral genome into the cytoplasm. The viral genome is composed of three negative sense RNA segments, termed small (S), medium (M), and large (L). The viral nucleocapsid protein (N) coats the genomic RNA segments and a single RNA-dependent RNA polymerase (L) molecule is attached to the complimentary ends of each segment [77]. N protein protects the viral genome and, upon release into the cytoplasm, the attached L synthesizes positive-sense RNA from the negative-sense genome. The viral mRNA is thought to be synthesized first, allowing for protein production to facilitate synthesis of the antigenomic and genomic RNA [78]. Whether there is a preference for the mRNA or antigenomic RNA to be made first is a topic of ongoing study.

Orthohantavirus replication and transcription both occur through a prime-and-realign mechanism, although how RNA synthesis is initiated differs between the two. Viral mRNA synthesis is initiated using a 5′ capped host-derived ~8–17 nt RNA molecule that serves to prime RNA synthesis [79,80]. This capped host RNA primer is obtained through a cap-snatching mechanism that is carried out by N and L [80,81]. N has been shown to preferentially bind capped nonsense RNAs [60]. In fact, transfection of N into HeLa cells resulted in increased abundance of 5′ termini of nonsense RNAs, suggesting that N protects the 5′ end of these RNAs from nonsense-mediated decay [81,82]. Evidence suggests that N binds specifically to the 5′ m7G cap and that this cap-binding region of the protein is distinct from its RNA-binding domain [81,83,84]. However, defining the structural region of N, which imparts this cap-binding activity, has remained a challenge. Several groups have presented crystal structures or high-resolution EM structures for N but were unable to identify a canonical cap-binding domain, leading to speculation that N may bind to the RNA itself as opposed to the 5′ cap structure specifically [85,86,87,88]. Following selection of capped host RNAs, N then associates with L, whose endonuclease activity cleaves the RNA to produce a host capped primer of ~8–17 nt [79,80]. L aligns the terminal G residue of the host RNA primer to the 3′-most cysteine on the genomic RNA and then adds the subsequent three nucleotides of the new positive sense mRNA strand [79,89,90,91]. The newly formed RNA then slips back three nucleotides and realigns to the 3′ end, with the initial guanine forming an overhang. The polymerase then elongates the nascent RNA strand until it reaches a transcription termination signal in the viral 5′ untranslated region (UTR) [92]. The transcription termination signal differs for each of the viral segments and varies among orthohantaviruses. For the S segment, the termination signal is associated with CCC-rich regions, with the specific sequence being a CCCACCC motif in the 5′ UTR of orthohantaviruses with reservoirs in the Sigmodontinae and Murinae subfamily of rodents [89]. For the M segment of SNV, transcription stops at a U8 polyadenylation-transcription termination signal, which results in the polyadenylation of the M segment mRNA. The L segment is not thought to have a transcription termination signal as sequencing of SNV L mRNA revealed that the mRNA was the same length as genomic L RNA. The presence of the transcription termination signals in the S and M segments results in the viral mRNA being shorter than the antigenomic and genomic RNA, with the difference in length varying between orthohantaviruses.

Synthesis of the antigenome, also called the complementary RNA (cRNA), is initiated when the polymerase aligns a guanidine triphosphate to the 3′–most cysteine on the genomic RNA molecule. Then, using the same prime-and-realign method described for mRNA synthesis, the polymerase produces the full-length antigenome. The 5′ overhanging guanine residue is cleaved, likely by the L protein, leaving a uridine monophosphate on the 5′ end [79,93,94]. After the antigenome is synthesized, it is then used as a template to make more genomic RNA, presumably through the same prime-and-realign mechanism [79]. Studies investigating genomic RNA replication of mosquito-borne bunyaviruses have shown that replication efficiency varies for each segment and is largely dictated by sequences in the 5′ and 3′ UTRs [95]. For Bunyamwera (BUNV), La Crosse (LACV), and Uukuniemi viruses (UUKV), the M segment genome was found to be the most abundant, followed by the L segment and the S segment genomes, respectively [95,96,97]. While a study characterizing SNV mRNA synthesis suggests that orthohantaviruses produce S, M, and L genomes at a ratio of 1:1:1, expansion of these experiments to other orthohantaviruses is needed to validate that this is true for orthohantaviruses as a whole [92].

The timing of RNA synthesis for each of the viral segments has been the focus of a number of investigations. In a study which measured the positive-sense viral RNAs (antigenome and viral mRNA together) produced during SNV infection of Vero cells, Hutchinson and colleagues observed the S RNA to be the most abundant, followed by the M RNA and then the L RNA [92]. The positive-sense S RNA was detected as early as 4 h post-infection, with the positive-sense M RNA detected at 8 h post-infection, and the positive-sense L RNA much later at 48 h post-infection. More recently, Wigren, Bystrom and colleagues quantified total viral RNA (positive- and negative-sense together) for each segment produced during PUUV infection of Vero cells [98]. They detected all three segments as early as 2 h post-infection and found that the production of viral RNA plateaued after 24 h post-infection. The S RNA was observed to be the most abundant at all time points measured, followed by the M RNA and the L RNA. Overall, these studies provide a solid baseline for future research to characterize the individual RNA kinetics of the genome, antigenome, and mRNA for all three orthohantavirus segments.

The cellular sites of orthohantavirus replication have thus far evaded identification, although concerted efforts to define the localization of viral proteins and RNA have offered a few theories. Membrane fractionation and immunofluorescence microscopy analyses of the New World orthohantaviruses, Black Creek Canal virus (BCCV) N in BHK21 cells and Tula virus (TULV) L in Vero cells demonstrated that both proteins localize to perinuclear regions and are membrane associated, indicating that this might be a site of transcription and RNA replication [99,100]. Investigations into stress granule formation during PUUV and ANDV infection of A549 cells revealed foci of viral RNA surrounding stress granules and P-bodies, suggesting that these may also be sites of viral replication [101]. It is possible that both the perinuclear region and areas associated with stress granule formation serve as sites of viral replication but represent different stages of RNA synthesis. As discussed above, stress granules are a source of capped-snatched host RNA. Therefore, the areas surrounding stress granules may represent proximal sites of viral mRNA transcription [81,82,102]. Furthermore, as interaction between the viral genome and glycoproteins is important for particle packaging, genome replication may occur at membrane-associated sites near the ER, where the glycoproteins are translated and processed [103,104,105]. Determining the locations of viral RNA synthesis and how the RNA is trafficked to these locations remains an active area of orthohantavirus research.

Production of viral RNA within the cytoplasm poses certain risks for orthohantaviruses. Host cells have evolved numerous mechanisms for viral RNA detection, including the RLRs RIG-I and MDA5. Several studies have shown that RIG-I is activated during HTNV infection of HUVEC and A549 cells [27,106,107]. Knocking out RIG-I resulted in reduced and/or delayed ISG activation in response to HTNV infection, while knocking out both RIG-I as well as MDA5 completely ablated ISG expression [27]. The results of these studies show that RIG-I, and to a lesser extent MDA5, are the primary sensors of HTNV infection in human cells. RIG-I is known to detect 5′ triphosphorylated RNAs as well as a number of other RNA sequences and structures [19,108,109,110,111,112,113,114,115]. Orthohantavirus genomic and antigenomic segments are complimentary at their 3′ and 5′ ends, allowing them to form short dsRNA panhandle regions that could theoretically be detected by RIG-I [79,116]. To avoid detection, orthohantaviruses trim the 5′ nucleotide of their genomic and antigenomic RNA during replication, as discussed above, leaving a monophosphorylated nucleotide that is less likely to activate RIG-I and downstream innate immune signaling [79,93,94]. However, the efficiency of genome trimming in infection is unknown. It is therefore possible that untrimmed 5′ triphosphorylated genomic RNAs are produced at some level sufficient to initiate RIG-I signaling. Although viral mRNA is synthesized with host cap structures, thereby evading RIG-I recognition, Lee and colleagues reported that mRNA for the N protein could be detected by RIG-I [106]. In this study, transfection of either expression plasmids for HTNV N or of in vitro transcribed RNA, containing a 5′ triphosphate moiety, into HEK 293T cells resulted in increased IFNβ promoter activity as determined by a luciferase reporter. Uncapped host RNAs (victims of cap-snatching) may also provide a source of stimulatory RNA for RIG-I activation in orthohantavirus infections. RIG-I detecting and binding of host RNAs during infection by KSHV, Influenza A virus, and HSV-1, occurring due to mislocalization of noncoding RNAs or abnormal transcript processing, has been reported [117,118,119]. Whether host RNA products from virally targeted cap-cleavage are capable of activating RLRs has not yet been investigated. Underscoring the importance of the RLR pathway in restricting virus infection, both pathogenic and nonpathogenic orthohantaviruses have evolved innate immune antagonism strategies against immune activation triggered by RIG-I. The N proteins of ANDV and TULV, the nonstructural (NS) proteins of ANDV, TULV, PUUV, PHV, SNV, and Khabarovsk virus (KHAV), the glycoprotein precursor (GPC) of PUUV, and the N-terminal glycoproteins (Gn) of ANDV, TULV, and NY-1 have all been reported to suppress signaling downstream of RIG-I and MDA5 [120,121,122,123,124,125]. Overall, these studies emphasize the importance of RIG-I recognition and activation to the orthohantavirus lifecycle and highlight how viral replication and the host innate immune response have influenced one another evolutionarily.

The activation of innate immunity leads to the production of antiviral factors that inhibit orthohantavirus RNA synthesis. MxA is a well-defined antiviral protein with activity against several negative-stranded RNA viruses [126,127,128,129,130,131]. When overexpressed in Vero cells, MxA inhibited the accumulation of viral transcripts and protein production in HTNV, PUUV, and TULV infection [128,132]. Interestingly, the antiviral activity of MxA may be cell type-dependent, as MxA expression did not impact PUUV or TULV replication in human monocytic U937 cells [132]. For the related bunyavirus LACV, MxA was shown to bind to and traffic the N protein into large perinuclear complexes [133,134]. This sequestration was hypothesized to inhibit N from coating the viral RNA, interfering with genome replication. Researchers have hypothesized that a similar mechanism may be involved in ANDV infection of HUVEC due to observed colocalization of MxA and N via IFA [135]. Overall, MxA has antiviral activity against both highly pathogenic and low-pathogenic orthohantaviruses, although possibly in a cell type specific manner. Research into the nature of the interaction between MxA and N will be important for defining the antiviral mechanism.

Another antiviral protein which has recently been shown to restrict orthohantavirus infection is ISG20 [76]. Overexpression of human ISG20 in Vero cells was shown to completely inhibit PUUV infection and to restrict several other bunyaviruses. The antiviral activity of ISG20 is attributed to its endonuclease activity as overexpression resulted in decreased levels of the genomic, antigenomic, and viral mRNA of the BUNV S segment. However, the exact mechanism of action and whether this is also true for orthohantavirus infection remains to be determined.

In summary, RIG-I has been shown to be an important sensor of orthohantavirus infection, being the primary detector of HTNV infection in human cells and creating evolutionary pressure on orthohantaviruses to develop means to avoid and/or inhibit RIG-I recognition and activation. Host cells can interfere with orthohantavirus RNA synthesis through the expression of MxA and ISG20. Remaining avenues of research include pinpointing where orthohantavirus RNA synthesis occurs within the cell, identifying the RNA PAMP that is being detected by RIG-I during infection, and the continuation of efforts to describe the antiviral mechanism of action for MxA and ISG20. Additional characterization of the replication kinetics for each orthohantavirus RNA species will also be important for determining the impacts of antiviral proteins on viral RNA synthesis at the molecular level.

### 2.3. Viral Translation

Although orthohantaviruses enter cells with nucleocapsid protein coating the viral RNA and a polymerase protein attached to each segment, they must synthesize new viral proteins to modulate the host cell immune response and metabolism as well as to generate progeny virions. During translation of host mRNA, the protein complex eukaryotic initiation factor 4F (eIF4F) and the polyA binding protein (PABP) associate with the 5′ cap and polyA tail [136]. Binding of this complex to the host mRNA is required for assembly of the large and small ribosomal subunits and the initiation of translation. Orthohantavirus mRNAs use a different strategy to initiate translation. Viral N mimics the functions of the eIF4F complex to facilitate translation of mRNA [84,137]. N binds to the 5′ end of the capped viral mRNA in association with the host 43S pre-initiation complex, loading the small ribosomal subunit onto the 5′ end of the RNA [84,138]. N also has RNA helicase activity, allowing it to dissociate any secondary RNA structures that might impede translation initiation [84]. Both the 5′ and 3′ untranslated regions (UTR) of the viral mRNA are reported to be involved in translation initiation [137,139,140]. N binds to the 5′ UAGUAGUAG sequence of orthohantavirus RNAs with high affinity, allowing N to preferentially recruit the host translation machinery to viral mRNAs [139]. Furthermore, evidence suggests that the 3′ UTR of S segment mRNA is able to enhance translation and functionally replaces the need for a polyA tail [137,140]. This is thought to involve interactions between the 3′UTR and 5′UTR, the N protein, and host proteins such as Mex3A, although the exact mechanism is unknown [137]. Whether the 3′UTR of the M and L segment mRNAs behave in a similar manner has yet to be determined.

An important antiviral mechanism inhibiting viral translation is through the activation of protein kinase R (PKR). PKR recognizes PAMPs such as dsRNA, and dimerizes and phosphorylates the eukaryotic initiation factor 2 (eIF2), effectively halting global translation initiation [141,142]. This results in the subsequent formation of stress granules, sequestering host translational machinery and further reducing protein synthesis [143]. While PKR is reported to restrict some bunyaviruses, several orthohantaviruses deploy mechanisms to inhibit PKR activation [144,145,146,147]. SNV, ANDV, and HTNV have all been shown to prevent PKR dimerization through association of the N protein with the PKR inhibitor P58IPK [146]. N strongly binds to P58IPK, outcompeting its inhibitor Hsp40, which forms an inactive complex with P58IPK. Dissociation of the P58IPK:Hsp40 complex by N allows P58IPK to bind directly to PKR, preventing PKR from being able to dimerize, thereby inhibiting its activity. An interesting, unresolved question is whether non-pathogenic orthohantaviruses antagonize PKR as well.

In addition to the phosphorylation of eIF2, another result of PKR activation is the formation of stress granules, which also inhibit translation. Interestingly, during PUUV and ANDV infection of HUVEC, PKR-dependent stress granules were found to form between 18 and 48 hpi, but not at earlier or later time points [101]. While HTNV infection in HUVEC did not induce significant stress granule formation, increased stress granules were observed in infection of A549 cells. These data indicate that orthohantavirus antagonism of PKR may not be efficient, resulting in transient stress granule formation. Furthermore, the viral RNA was not found to colocalize within these stress granules, but was found in close proximity to them. Thus, the viral RNA may avoid sequestration in stress granules. Christ et al. suggest that areas of viral RNA accumulation around stress granules may be sites of viral replication and/or translation. Overall, research suggests that orthohantaviruses strike a balance between PKR activation and inhibition, which allows for efficient viral replication and translation.

In summary, orthohantaviruses largely inhibit PKR activation, preventing it from halting global translation through a well-described mechanism. However, transient activation of PKR has been shown to occur during infection, leading to stress granule formation, which may play a role in orthohantavirus replication. Continued research identifying other host antiviral proteins that interfere with orthohantavirus translation will be important for addressing the interplay between innate immunity and translation.

### 2.4. Virion Assembly and Release

Virus particle assembly and budding are thought to occur at tubular projections within the Golgi and/or at the plasma membrane, although which serves as the primary site of particle budding and formation is debated [148,149,150,151,152,153,154]. Old World orthohantaviruses are generally thought to assemble in the Golgi [153,154]. Recently, however, Parvate and colleagues used cryofixation of HTNV-infected Vero cells followed by electron microscopy to visualize extracellular virions in close proximity to projections on the plasma membrane, suggesting budding at the cell surface [149]. In contrast, virion assembly for the New World orthohantaviruses is thought to occur at the plasma membrane, although studies using electron microscopy and glycoprotein localization have shown evidence for assembly at internal membranes as well [150,151,152].

The orthohantavirus glycoproteins are co-translationally cleaved at a conserved WAASA site into the individual Gn and Gc proteins [103]. These proteins are then glycosylated, form trimeric complexes in the ER consisting of a Gn dimer and a Gc monomer, and are then trafficked to the Golgi [153,155,156,157,158,159]. Studies have shown that each glycoprotein can localize to the Golgi when expressed independently and that co-expression of both proteins greatly enhances their trafficking from the ER to the Golgi [104,150,154,160,161,162]. Once at the Golgi, glycoprotein spikes are assembled from the trafficked complexes and accumulate at internal membranes [153]. Expression of Gn and Gc is sufficient for virus-like particle formation, as transfection of cells with expression plasmids for either ANDV GnGc or PUUV GnGc resulted in the production of virus-like particles [163].

Packaging of the viral genomic RNA is thought to be mediated by interactions between the N protein, the viral RNA, and zinc finger domains in the cytoplasmic tail of Gn [164,165,166,167,168]. The mechanisms that ensure efficient packaging of all three genomic segments into orthohantavirus particles are still unknown. The 5′ and 3′ UTRs, which allow for panhandle formation and circularization of the RNA, are known to be important for viral packaging of other bunyaviruses [169,170]. Studies utilizing mini-replicon reporters for UUKV and BUNV observed that L genomic RNA is most efficiently packaged into particles, followed by M and S [171,172]. Furthermore, Bermudez-Mendez et al. quantified packaging efficiency of bunyaviruses Rift Valley Fever virus (RVFV) and Schmallenberg virus (SBV) and found that less than 10% of viral particles produced during mammalian cell infection contained all three viral segments, with many particles containing only one or two viral segments [173]. Surprisingly, up to 55% of viral particles produced contained no genomic RNA. This, along with other studies that report most mosquito-borne bunyavirus particles do not contain all three viral segments, leads to the hypothesis that packaging of bunyavirus RNA is not segment-specific and that productive infection may rely on co-infection, or complementation of multiple incomplete particles [174,175,176]. How efficiently each orthohantavirus segment is packaged and whether a similar method may be used in orthohantaviruses remains to be elucidated.

Currently, no host innate immune antiviral proteins have been identified to inhibit particle assembly and release of orthohantaviruses. Clues may lie in those reported for other segmented viruses. In particular, viperin is reported to inhibit the arenavirus Junin virus and influenza A virus (IAV) particle assembly by disrupting the formation of lipid rafts, which serve as a site of viral budding on the plasma membrane [177,178]. Tetherin was also identified to inhibit release of arenaviruses and IAV by retaining viral particles at the cell membrane [179,180,181,182]. Interestingly, tetherin correlates with the host range for several bunyaviruses, in that human tetherin restricted the infection of viruses with ruminant tropism and, likewise, sheep tetherin restricted viruses with human tropism [183]. If tetherin inhibits orthohantavirus infection and whether it plays a role in determining the rodent reservoir has yet to be determined. Overall, how the innate immune response detects and inhibits particle assembly and release represents a gap of knowledge and an ongoing area of research in the orthohantavirus field.

## 3. Challenges and Future Goals

A significant challenge to orthohantavirus research is the limited availability of molecular tools with which to investigate viral infection. More specifically, the development of tissue culture models for reservoir species, tractable reverse genetics systems, and strand-specific assays to measure viral RNA replication would allow the orthohantavirus field to better define the molecular interactions and mechanisms that determine efficient infection.

Reservoir tissue culture models: The vast majority of molecular orthohantavirus research has been done using human and non-human primate tissue culture models. While these models are important for understanding the interplay between viral replication and the innate immune response in the context of species which develop disease, they give an incomplete picture of the ecologically important host–pathogen interactions. Orthohantavirus infections in rodent reservoirs are asymptomatic, resulting in mild immune responses that fail to clear the virus but also do not elicit immunopathology [1,2,3]. This suggests unique mechanisms of immune modulation that are host-specific and potentially co-evolved. Further, viral replication may be differentially regulated in reservoir cells in order to prevent the generation or accumulation of PAMPs, thereby evading immune activation. Additionally, propagation of orthohantaviruses in non-reservoir models, such as Vero cells, has led to adaptions in the virus for these cell types that do not reflect what is present in the original virus [184]. Currently, established tissue culture models of both human and rodent reservoir models exist for SEOV, PUUV, and TULV [2,185,186]. While access to host and reservoir cell lines for these Old World orthohantaviruses allows investigation of previously posed questions, technical barriers still exist. First, while rat cell lines are well characterized as a laboratory model for genetic disease, vascular physiology, and aging, there are still fewer molecular tools available for rat when compared to human or mouse cell lines. This issue is exaggerated for vole cell lines, for which few molecular tools exist. Furthermore, as has been shown throughout this review, not all orthohantaviruses interact with host cells in the same way. Therefore, the generation of validated cell lines for multiple reservoir species is one step that would allow for better understanding of orthohantavirus replication and immune interactions.

Strand-specific qRT-PCR: Orthohantaviruses produce multiple RNAs that have specific roles during infection. However, there is no published assay to measure the genome, antigenome, and viral mRNA of each of the orthohantavirus segments individually. Strand-specific qRT-PCR strategies have been developed for viruses of several families. These assays rely on sense-specificity and unique features of individual RNA species, such as the presence of a 5′ cap or poly-A tail, to differentiate the viral RNAs [187,188,189,190,191]. Developing such an assay for orthohantaviruses using similar strategies would allow for a detailed understanding of the replication kinetics for each of the viral RNAs. This would provide invaluable insights into orthohantavirus infection such as whether the mRNA, antigenome, and genome of each segment are made in different stages or all simultaneously. Being able to compare individual RNA kinetics between different infection conditions would also allow researchers to define the molecular mechanisms of RNA replication and the individual requirements that determine whether the viral polymerase synthesizes the genome, antigenome, or mRNA. In particular, understanding the relative amounts and timing of viral RNA production will be especially important for the development and assessment of antivirals that target orthohantavirus RNA synthesis. Thus, the creation of a strand-specific assay for RNA quantitation would greatly benefit the orthohantavirus field.

Reverse genetics: The ability to manipulate the viral genome via mutation or the insertion of reporters is a powerful tool with which to interrogate the molecular mechanisms of viral replication and host interaction. For many viruses, this is accomplished through the creation of infectious clones encoded in DNA plasmids that allow researchers to easily alter viral sequences. Minigenome systems consist of plasmids encoding the 5′ and 3′ UTRs of the viral genome, which flank reporter genes such as luciferase or GFP. Minigenomes do not produce fully infectious virus but are useful for studying the role of the UTRs in viral RNA synthesis, the packaging of RNA into particles, and RNA-protein interactions [95,171,172,192,193]. Because of the incredible usefulness of these systems for molecular virology, concerted efforts have been made to develop reverse genetics systems for orthohantaviruses, resulting in minigenome systems being proposed for HTNV and ANDV [194,195]. However, these systems lack universal tractability and have not yet been widely adopted. One possible explanation for the challenges of a reverse genetics system for orthohantaviruses may be the relatively narrow range of cell lines that support efficient viral replication. It is likely that the cell lines used to rescue minigenome systems or replicate cDNA based virus genomes may not be ideally suited for low levels of orthohantavirus replication. Although few reservoir cell lines currently exist, it may be reasonable to hypothesize that recombinant virus rescue may be more robust within these well-adapted cells [3]. The development of reverse genetics systems for the mosquito-borne bunyaviruses ushered in huge advances in understanding how these viruses replicate and interact with the host [95,171,172,192,193,196]. As such, the generation of a cDNA-based reverse genetics system remains a top priority for the orthohantavirus field.

## 4. Conclusions

Finally, it remains critical to define the spectrum of innate immune genes and ISGs that impart antiviral actions to suppress orthohantavirus infection, replication, and cell-to-cell spread. What are the genes that control infections that otherwise lead to HFRS and HCPS, and how do innate immune activation, IFN, and immune cytokines regulate disease outcome? Defining specific factors and their mechanisms of action that mediate the innate immune control of orthohantavirus infection will inform the consideration of therapeutic strategies to leverage specific innate immune actions of host cells for the mitigation of infection and disease.

## Data Availability

Not applicable.

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
