# Peer review of "Orthohantavirus Replication in the Context of Innate Immunity"

_viruses, 2023, doi:10.3390/v15051130_

Round 1

Reviewer 1 Report

In their manuscript, the authors review the current knowledge on the interactions that hantaviruses have with the innate immunity of their hosts during the different stages of the viral cycle (entry, replication, and release of viral particles). Then the authors point to the challenges to be met to better understand hantavirus replication, in a context of lack of genetic, molecular, and cellular tools.

The manuscript is well documented, and the topic is of importance. However, the general description of the viral cycle of hantaviruses, already widely reviewed in the literature, appears too detailed in relation to the question addressed by the authors. For instance, Meier et al., Viruses, 2021, although with a different perspective, have presented in the same order, literature data on hantavirus entry, genome replication, transcription, assembly and egress.

Furthermore, the organization of the manuscript makes it somewhat difficult for the reader to easily get an idea of the progress made in the field of innate immunity in relation to the different stages of hantavirus replication. This should be improved in part 2 of the manuscript, by the addition at the end of each section (2.1 to 2.4) of a short conclusion on the related activation and antagonism properties of hantaviruses at each stage of the viral cycle. For instance, the conclusions in section 2.1: line 98 and line 144, could be moved at the end of this section.

-       Section 2.2 overemphasizes information on replication and transcription of viral genomes while there is fewer comments on the link to innate immunity. Lines 278-285: information does not reflect all that is known about the role of different viral proteins from different hantaviruses, in particular antagonism of RIG-I signaling has not been described only for the NSs of ANDV (besides NSs is not a feature of all hantavirus species).  Also, the role of N and GPC of different hantaviruses individually expressed and not only “N-terminal glycoprotein” (not clear: does the authors referred to cytosolic domain of Gn) have been studied.

-       Section 2.3 is better balanced between general data on translation and its inhibition by hantaviruses.  Explanation of how N proteins from different hantaviruses inhibit the antiviral activity of PKR is not clear and could be improve (line 336-339) : is it that the N proteins of SNV, ANDV and HTNV prevent the dimerization of PKR and therefore its activation, by allowing the binding of P581K to PKR.  For this, N competes with Hsp40 for its binding to P581K, thus releasing the latter from its inactive form when associated to Hsp40…..

-       Section 2.4:  line 362: ref (143) by Xu et al is not appropriate to illustrate Parvate’s work and should not appear here, or the sentence should be modified. Line 369 and 372: The work of Serris, ref (154) relates to structural analyses of the lattice formed by Gn and Gc on the surface of viral particles, and to the role of Gc in the fusion process during entry. The link with glycoprotein trafficking to the Golgi is not clear. Line 370: that both Gn and Gc are retained in the ER when individually expressed should be mitigated (e.g., Sperber’ref 149 indicates a localization of Gn in the ER and Gc in the Golgi).

-       In part 3 of the manuscript, to illustrate reservoir tissue culture models, the authors should include other immortalized cell lines (Eckerle et al, Viruses, 2014, Binder et al, J Virol Methods, 2019) allowing the study of other hantaviruses than only SEOV and PUUV.

In the introduction line 27 and 62 references (3) and (26) correspond to the same publication (see line 494 and 534 of the reference list)

For reading fluency, reference repetitions in neighboring sentences could be removed.

 ref (89) lines198, 200, 203;

ref (103) lines 270, 273;

ref (128, 129) lines 293, 295

ref (141) lines 338, 339

ref (98) lines 344, 346, 349, 351

and so on…

References should be listed in the format of Viruses by indicating names of the authors rather than only the first author.

In conclusion the subject presented by the authors is of great interest to proceed in hantavirus knowledge. Some improvement of the manuscript and greater attention to the referenced literature are nevertheless necessary.

Reviewer 2 Report

In this manuscript, the authors provide an overview of the current knowledge on orthohantavirus replication and the effects of replication on activation of the host innate immune system. It is known that some pathogenic orthohantaviruses are able to evade the host immune response at multiple levels, but the link to the development of HFRS or HCPS is not entirely clear.

In the manuscript, the authors describe in a very detailed and organised manner the current knowledge on the attachment and entry of various pathogenic orthohantaviruses into the host cell, viral RNA replication and transcription, viral translation, and virion assembly and release.

Activation of innate immunity by orthohantaviruses and the host antiviral response are an important part of orthohantavirus pathogenesis that is not fully understood. The present manuscript is important to the field in this regard. I have some minor comments or suggestions for the authors.

1. The manuscript is well and concisely written and provides the reader with a wealth of concise information. Additional schematic representation of viral replication and innate immune responses would be useful for better visualization and understanding.

2. In the manuscript, the authors use the term Hantavirus. Instead of Orthohantavirus, which is a new name for the genus (ICTV). In the field of these viruses, the name Hantavirus has come into common usage. However, I believe that it would be more appropriate to use the new name in the manuscript.

3. It would also be interesting to highlight the differences between pathogenic and nonpathogenic orthohantaviruses at specific points in the immune response when data are available for nonpathogenic orthohantaviruses (e.g., the results of in vitro experiments in which nonpathogenic orthohantaviruses were included as controls).
